# Effect of Prehospital Epinephrine Use on Survival from Out-of-Hospital Cardiac Arrest and on Emergency Medical Services

**DOI:** 10.3390/jcm11010190

**Published:** 2021-12-30

**Authors:** Song Yi Park, Daesung Lim, Seong Chun Kim, Ji Ho Ryu, Yong Hwan Kim, Byungho Choi, Sun Hyu Kim

**Affiliations:** 1Department of Emergency Medicine, Dong-A University College of Medicine, Dong-A University Hospital, Busan 48114, Korea; capesong@naver.com; 2Department of Emergency Medicine, Gyeongsang National University College of Medicine, Gyeongsang National University Changwon Hospital, Changwon 51472, Korea; daesung2@gmail.com (D.L.); gsimem@naver.com (S.C.K.); 3Department of Emergency Medicine, Pusan National University College of Medicine, Pusan National University Yangsan Hospital, Busan 50612, Korea; pnuyhem@gmail.com; 4Department of Emergency Medicine, Samsung Changwon Hospital, Sungkyunkwan University School of Medicine, Changwon 51353, Korea; suka1212@naver.com; 5Department of Emergency Medicine, University of Ulsan College of Medicine, Ulsan University Hospital, Ulsan 44033, Korea; byungho2000@gmail.com

**Keywords:** out-of-hospital cardiac arrest, emergency medical service, epinephrine, survival rate

## Abstract

This study was to identify the effect of epinephrine on the survival of out-of-hospital cardiac arrest (OHCA) patients and changes in prehospital emergency medical services (EMSs) after the introduction of prehospital epinephrine use by EMS providers. This was a retrospective observational study comparing two groups (epinephrine group and norepinephrine group). We used propensity score matching of the two groups and identified the association between outcome variables regarding survival and epinephrine use, controlling for confounding factors. The epinephrine group was 339 patients of a total 1943 study population. The survival-to-discharge rate and OR (95% CI) of the epinephrine group were 5.0% (*p* = 0.215) and 0.72 (0.43–1.21) in the total patient population and 4.7% (*p* = 0.699) and 1.15 (0.55–2.43) in the 1:1 propensity-matched population. The epinephrine group received more mechanical chest compression and had longer EMS response times and scene times than the norepinephrine group. Mechanical chest compression was a negative prognostic factor for survival to discharge and favorable neurological outcomes in the epinephrine group. The introduction of prehospital epinephrine use in OHCA patients yielded no evidence of improvement in survival to discharge and favorable neurological outcomes and adversely affected the practice of EMS providers, exacerbating the factors negatively associated with survival from OHCA.

## 1. Introduction

Out-of-hospital cardiac arrest (OHCA) is a serious public health concern. The survival rate for OHCA patients varies according to the emergency medical services (EMSs) in each country. However, the dramatic improvement of survival rate over the decades does not seem to have been significant [1,2]. There have been many prehospital efforts to increase the survival rate of OHCA patients, and current guidelines recommend epinephrine use for advanced life support (ALS) in OHCA [3,4].

There are several studies on the use of epinephrine at the prehospital stage. A study in Singapore reported that the introduction of intravenous epinephrine to an EMS system did not yield a significant survival benefit [5]. A study in Japan reported that prehospital administration of epinephrine by an EMS is favorably associated with long-term neurological outcomes in patients with initial asystole [6]. However, a recent study by Perkins et al. [7] showed that prehospital use of epinephrine resulted in a significantly higher chance of survival but no significant difference in neurological outcome.

Prehospital epinephrine use is one of the efforts to improve survival in OHCA patients. However, countries have different prehospital EMS systems [8,9,10]. Even within the same EMS system, there may be differences in the capabilities of EMS providers in each region [11]. Therefore, the prehospital effect of epinephrine needs to be considered along with factors related to OHCA survival within the EMS system [12,13]. However, studies thus far seem to have focused only on whether epinephrine improves the survival rate in patients with OHCA.

If epinephrine is introduced for the first time in the prehospital stage, it is necessary to verify the effects of its intervention not only on the survival of OHCA patients but also on the impact of the EMS system. Therefore, the purpose of this study was to identify the effect of epinephrine on the survival of OHCA patients and changes in the prehospital EMS after the introduction of prehospital epinephrine use by EMS providers.

## 2. Methods

### 2.1. Study Design

This was a retrospective observational cohort study. This study reanalyzed the data of a previous investigation, as it coincided with the time when EMS providers started using epinephrine [14]. The previous study used the survival of OHCA patients before and after COVID-19 as a variable, and the present study analyzed survival according to the use of epinephrine as a variable.

This study compared the survival outcome and prehospital EMS changes in the two groups (epinephrine group and norepinephrine group) among OHCA patients attended by EMS providers after the introduction of epinephrine. All patients for whom resuscitation was attempted by EMS providers due to OHCA during the study periods were included in the study. Patients were excluded if they were less than 18 years old and had a valid do-not-resuscitate order. Patients were also excluded if the cause of arrest was presumed to be trauma, drowning, or intoxication. The first data collection period was from 1 November 2019 to 31 January 2020, and the secondary data collection period was from 1 November 2020 to 31 January 2021.

### 2.2. Study Setting

The Busan, Ulsan, Gyeongnam, and Changwon regions, where this study was conducted, are located along the coast in the southeastern part of Korea. This region consists of two metropolitan cities (Busan and Ulsan, Korea), one city (Changwon, Korea), and one province (Gyeongnam, Korea), with a total population of 792 million, spread over almost 12,369 km^2^ [15].

The EMS system in Korea, which is government-based and single-tiered, provides basic to intermediate levels of EMSs, such as supraglottic airway insertion, tracheal intubation, and basic life support from fire agency headquarters. The EMS resuscitation protocol introduces multiple dispatches (two or more ambulance teams), provides on-site cardiopulmonary resuscitation (CPR), and transports patients to an emergency department (ED) in an ambulance with ongoing CPR. EMS providers cannot stop CPR unless the patient exhibits return of spontaneous circulation (ROSC), either on-site or during transportation to the ED, and only physicians in hospital EDs can declare death [16]. Most EMS teams in urban areas consist of three EMS providers, usually including at least one emergency medical technician. Many of the EMS providers have the certification of registered nurses or first/second-grade emergency medical technicians. Ambulances staffed with a physician are not available. Among the practices of EMS providers, advanced airway management, intravenous access, and fluid administration, and withholding/withdrawal of resuscitation are performed under medical oversight by medical directors, who are mostly emergency physicians [17].

As of November 2019, prehospital use of epinephrine by EMS providers was permitted in the study region as part of the national pilot project. Certified EMS providers can administer epinephrine under the video-medical oversight of medical directors after three days of training. The regional guidelines for the use of epinephrine in OHCA patients were made by medical directors in the region and were as follows: (1) administration of epinephrine is permitted to be performed only by a certified EMS team and under the video-medical oversight of a medical director, (2) epinephrine should be administered at 1 mg every 4 min, (3) in multiple dispatches, whether the team is a certified EMS team or not, the on-scene stay should not exceed 15 min, (4) do not use epinephrine in the ambulance during patient transport, but focus on high-quality chest compressions. This guideline was prepared in consideration of the local EMS provider’s proficiency, human resources, access to the ED, and research results to date [18]. As of November 2019, not all EMS providers in the region were certified for prehospital use of epinephrine. As a result, some patients received epinephrine, and some did not.

However, since the first confirmed case of coronavirus-19 (COVID-19) was reported in Busan on 21 February 2020, there have been some small changes in the practice of EMS providers to reduce exposure to COVID-19. It was recommended that personal protective equipment (PPE) be put on before entering a scene, that the number of dispatched personnel be limited, and that the use of mechanical chest compression devices be considered. It was also recommended that high-efficiency particulate air filter respirators be used for all ventilation procedures and that an EMS provider intubates with the highest chance of first-pass success and, if intubation was delayed, to consider the use of a supraglottic airway according to the guidelines [19].

In the ED, there are no significant differences in the ALS protocol for OHCA patients. Except when the patient has no clear evidence of death or is medically futile for resuscitation, the emergency physician performs the ALS for at least 20 min and then decides whether to stop. If the patient is resuscitated and has the appropriate indications, targeted temperature management (TTM) is performed in most cases [20].

### 2.3. Data Sources

The study region has not yet been equipped with an integrated cardiac arrest registration system. Therefore, data from the prehospital and hospital stages were collected, matched, and merged. Prehospital data on all dispatches of EMSs are collected and managed by regional fire agencies electronically from scene-dispatched EMS providers. For cases where resuscitation is performed, the EMS providers file a prehospital cardiac arrest patient care report. In this study, anonymous prehospital data were collected from the four headquarters of the national fire agencies. Hospital data were collected from treating hospital EDs (76 EDs from the first data collection period and 69 EDs from the second data collection period).

### 2.4. Variables and Measurements

Data for the patient variables of age, sex, and medical history, including a history of hypertension, diabetes, stroke, cardiac disease, pulmonary disease, liver disease, renal disease, and malignancy were collected. Data for bystander variables, bystander-witnessed status, and bystander CPR status were collected.

Data for the EMS variables of the presence of an initial shockable rhythm on the scene, the use of advanced airway management (I-gel/supraglottic airway, tracheal intubation, or no advanced airway management), the use of mechanical chest compression devices, the use of epinephrine, and EMS process time (response, scene, and transport times) were collected. EMS response, scene, and transport times were defined as the time elapsed from the call to EMS arrival at the scene, from EMS arrival at the scene to EMS departure from the scene, and from EMS departure from the scene to EMS arrival at the ED, respectively.

Data for the hospital variables of ROSC at any time, survival to discharge, and favorable neurological outcome were collected. ROSC at any time was defined as whether there was a pulse at any time during hospital CPR regardless of the patient’s survival status. Survival to discharge was defined as the case in which a patient survived until hospital discharge. Favorable neurological outcome was defined as cerebral performance category (CPC) 1 or 2, among the following categories: CPC 1 (good cerebral performance), CPC 2 (moderate cerebral disability), CPC 3 (severe cerebral disability), CPC 4 (coma or vegetative state), and CPC 5 (dead) [21].

### 2.5. Outcome Measures

The primary outcome of the study was the survival to discharge and favorable neurological outcomes of OHCA patients. The secondary outcome was the change in the prehospital EMS after the introduction of prehospital epinephrine use by EMS providers.

### 2.6. Statistical Analysis

We performed a descriptive analysis to examine the distribution of variables. Continuous variables are presented as the mean and standard deviation (SD) or median and interquartile range (IQR), and categorical variables are presented as frequencies and proportions. In the comparison of the two groups, we compared the group differences using independent *t*-tests or Mann-Whitney U tests for numeric variables and chi-square tests or Fisher’s exact tests for categorical variables. Differences in patients’ baseline characteristics were observed when a direct comparison of outcomes was performed. We used propensity matching to approximate a randomized trial to reduce the influence of selection. We considered all variables, including nonsignificant variables, to form a propensity model. From this, a propensity score was generated for each patient from a logistic regression model based on 18 confounding variables. An 8-to-1 digit greedy matching algorithm was used to identify a unique matched control for each epinephrine patient according to the propensity score using the SAS system PROC LOGISTIC, and a greedy match algorithm was used to match cases to controls (SAS macro code: http://www2.sas.com/proceedings/sugi29/165-29.pdf, accessed on 23 December 2021) [22,23]. Under this algorithm, if this match could not be found, the algorithm then proceeded sequentially to the next highest digit match on the propensity score to make “next-best” matches in a hierarchical sequence until no more matches could be found. Once a match was made, the match was not considered again. For 1:1 matched data, we compared the group differences using the paired *t*-test or Wilcoxon’s signed-rank test for numeric variables and the McNemar test for categorical variables. We identified the association between outcome variables regarding survival and epinephrine controlling for confounding factors based on univariate and multivariate conditional logistic regression analysis with odds ratios (ORs) and 95% confidence intervals (CIs). All statistical analyses were performed using SAS 9.4 (SAS Institute Inc., Cary, NC, USA). A two-sided *p*-value of <0.05 was considered statistically significant.

### 2.7. Ethical Statement

This study was reviewed and approved by the Institutional Review Board of Dong-A University Hospital (approval no. DAUHIRB-EXP-21-221). The requirement for informed consent from patients was waived because the study was a retrospective analysis of existing data that did not contain personal patient information at the time the data were provided.

## 3. Results

A total of 2884 OHCA patients were eligible during the study period. Among them, 588 patients were excluded by exclusion criteria, and 59 patients were excluded by events that occurred in health care facility staffing with physicians. In the data curation, 248 patients were excluded due to duplicated data, missing in-hospital data, refusal to provide hospital data and the absence of data on epinephrine. Finally, 1949 patients were included in the study population; 337 were classified as the epinephrine group, and 1612 were classified as the norepinephrine group. The flow chart of the study population is shown in Figure 1.

Characteristics of the total study population and 1:1 propensity-matched population.

In the total study population, there were significant differences between the epinephrine and norepinephrine groups in sex, medical history of liver disease, bystander CPR, initial rhythm, use of a mechanical chest compression device, and EMS process time. The characteristics of the total study population and the 1:1 propensity-matched population are shown in Table 1.

The outcomes of prehospital epinephrine administration by EMS providers during OHCA on survival to discharge and favorable neurological outcomes.

The survival-to-discharge rate and OR (95% CI) in the epinephrine group were 5.0% (*p* = 0.215) and 0.72 (0.43–1.21), respectively, in the total patient population and 4.7% (*p* = 0.699) and 1.15 (0.55–2.43), respectively, in the 1:1 propensity matched population (Table 2). The favorable neurological outcomes were 3.6% (*p* = 0.218) and 0.68 (0.37–1.26) in the total patient population and 3.1% (*p* = 0.816) and 1.11 (0.45–2.73) in the 1:1 propensity score matching population (Table 2). These results were consistent when matching was expanded to 1:N (Appendix A).

In the subgroup analysis according to the initial rhythm, survival to discharge was 1.8% (*p* = 1.000) and 1.00 (0.29–3.45) in the non-shockable rhythm group and 23.8% (*p* = 0.595) and 2.000 (0.1–22.06) in the shockable rhythm group in the 1:1 propensity matched population (Table 3).

The changes in EMS after the introduction of prehospital epinephrine administration by EMS providers.

During the study period, the epinephrine group received more mechanical chest compression (66.2% versus 32.2%, *p* < 0.0001) and had longer EMS response times (9.62 ± 5.78 versus 8.88 ± 6.18, *p* < 0.017) and scene times (18.24 ± 5.87 versus 13.57 ± 6.25, *p* < 0.0001) than the norepinephrine group (Table 1). Mechanical chest compression was a negative prognostic factor for survival to discharge (0.25, 0.14–0.11) and favorable neurological outcomes (0.20, 0.09–0.42) in the epinephrine group (Table 4 and Table 5).

## 4. Discussion

This study aimed to identify the effect of prehospital epinephrine use by EMS providers during OHCA on patient survival to discharge and favorable neurological outcomes and prehospital EMSs in a region. There was no significant improvement in either survival to discharge (1.15, 0.55–2.43) or favorable neurological outcomes (1.11, 0.45–2.73) in the epinephrine group. After the introduction of epinephrine, the EMS provider used more mechanical chest compression devices, and the EMS response and scene time increased in the epinephrine group. Although this study is not large in scale, it has the strengths of analyzing the effectiveness of epinephrine by carefully examining and matching many prehospital confounding factors that affect the survival of OHCA patients and focusing on tracking the prehospital response of EMS providers after a new intervention.

Our findings are partially consistent with a previous observational study. In that study (conducted in Japan) in which the context of the EMS was similar, the use of prehospital epinephrine was associated with a decreased chance of survival and good functional outcomes one month after the event, even though there was an increased chance of ROSC before hospital arrival [24]. The difference between the two studies is that the present study more thoroughly identified and analyzed variables that could affect the survival of OHCA patients, including underlying disease and EMS provider practice (use or nonuse of a mechanical chest compression device, airway types, scene time), than previous studies. However, our study showed no statistically significant improvement in survival in ROSC at any time despite the well-matched group. This result is estimated to be due to the EMS response time (9.62 ± 5.78) of the epinephrine group in this study is longer than that (7.54 ± 4.0) in the previous study. There are many slope residences in Busan that are difficult to access by ambulance due to narrow pathways, so the response time may be delayed. However, the response time in the epinephrine group was no longer than that in the norepinephrine group (9.62 ± 5.78 versus 8.88 ± 6.18, *p* = 0.017). This difference may be the time taken to recruit the certified EMS team. During the study period, EMS providers were divided into teams that could or could not use prehospital epinephrine according to their certification status, and consequently, dispatchers would have to find an available team.

A study by Perkins et al. showed that the use of epinephrine resulted in a significantly higher rate of 30-day survival than the use of placebo (3.2% in the epinephrine group and 2.4% in the placebo group, 1.39, 1.06–1.82) [7]. However, our study did not show any improvement in survival to discharge (5.0% in the epinephrine group and 6.9% in the norepinephrine group, 0.72, 0.43–1.21). Factors contributing to such a difference might include the difference in the proportion of witnesses of cardiac arrest. The proportion of witnessed cardiac arrest in the previous study was 61.4% in the epinephrine group and 61.0% in the placebo group. However, in our study, it was 38.9% in the epinephrine group and 42.6% in the norepinephrine group. This was because our study included a patient cohort during the COVID-19 period. Another factor that might have contributed to different results between the present study and the PARAMEDIC2 study was that ALS was performed sufficiently at the scene, with a mean time interval between ambulance arrival at scene and departure of 50.1 ± 21.8 min in the epinephrine group and 44.5 ± 18.3 min in the placebo group. However, in our study, we recommended not to stay at the scene for more than 15 min. In addition, epinephrine was not used in the ambulance during transportation.

In a study of prehospital epinephrine use and EMS CPR duration, which was defined as the interval from the initiation of CPR by EMS staff to the arrival at the hospital or ROSC, epinephrine use was associated with an increased ROSC and improved neurological outcome when the CPR duration of EMS was between 15 and 19 min (1.327, 1.017–1.733) [25]. A CPR duration of 20–26 min was negatively associated with a favorable neurological outcome (0.967, 0.774–1.207). However, in our study, the CPR duration was much longer in both the epinephrine group (26.53 ± 15.16) and the norepinephrine group (23.24 ± 17.62). The reason for the longer CPR duration in the epinephrine group was due to the longer scene time in the epinephrine group than in the norepinephrine group (18.24 ± 5.87 versus 13.57 ± 6.25, *p* < 0.0001). Regional guidelines recommended not staying on the scene for more than 15 min; however, it seems that these guidelines were not followed well in the epinephrine group. In this regard, further investigation and studies are needed.

In this study, EMS providers used a mechanical chest compression device more often than performed manual chest compression in the epinephrine group (66.2% versus 32.2%, *p* < 0.001). However, mechanical chest compression was a negative prognostic factor for survival to discharge (0.25, 014–0.44) and favorable neurological outcomes (0.20, 0.09–0.42) of OHCA patients in the epinephrine group. From these findings, we have two questions: (1) why were mechanical chest compression devices were used more often in the epinephrine group than in the norepinephrine group? (2) was the quality of mechanical chest compression inferior to that of manual chest compression? Our estimation for the first is that with the current situation, three EMS providers were dispatched to one ambulance (with one of them driving), the introduction of prehospital epinephrine use was believed to have caused a shortage of labor to apply manual chest compression, although EMS resuscitation protocol introduced multiple dispatches (two or more ambulance teams) as described in the study setting. Regarding the second question, EMS providers in the region use LUCAS, Easy Pulse, and AutoPulse as mechanical chest compression devices. However, equipment that can monitor and control the quality of CPR including capnometry and CPR quality improvement programs is not routinely used. In an animal study comparing LUCAS and manual chest compression, the use of mechanical devices provided ongoing high-quality chest compression and tissue perfusion during ambulance transport [26]. Another study showed that patients in the mechanical CPR group had a higher chest compression fraction than those in the manual CPR group (0.84, 0.78–0.91) [27]. However, some studies have indicated no evidence that the use of mechanical chest compression devices is associated with an improved survival rate of OHCA patients or poor neurological outcomes at hospital discharge [28,29,30]. The second question requires in-depth investigation and further research based on regional data.

In the subgroup analysis according to the initial rhythm, prehospital epinephrine use for OHCA patients was not associated with improved survival to discharge or favorable neurological outcomes, and this was not only in the non-shockable rhythm group but also in the shockable rhythm group. However, Tomio et al. [6] reported that prehospital epinephrine administration by EMS is favorably associated with long-term neurological outcomes in cases of non-shockable rhythm. The result of our study is thought to be due to the small number of samples.

Several limitations of our study must be acknowledged. First, detailed in-hospital treatment of post–cardiac-arrest care, including TTM, was not considered. However, the ED protocol of post–cardiac-arrest care in the region is not much different, as described in the study setting. Many EDs conduct TTM, and if it is not available, the patient is transferred to another available ED. Second, the frequency of epinephrine use in prehospital care was not analyzed. The number of epinephrine doses was related to the survival outcome [31]. According to regional guidelines, EMS providers are not recommended to stay on scene for more than 15 min, and epinephrine can be administered at 1 mg every 4 min. Thus, the maximum frequency of epinephrine administration will be 3 times per duration on scene. However, in this study, an analysis of the survival rate according to the frequency of epinephrine use was not performed. Third, mechanical chest compression devices were more commonly used in the epinephrine group, and the use of mechanical devices was related to poor prognosis in this study. The use of mechanical devices may be related to chest compression quality. However, in this study, the quality of chest compression was not investigated. Fourth, this study reanalyzed the data of a previous study to identify the EMS response before and during the COVID-19 pandemic [20]. According to a previous study, during the COVID-19 pandemic, tracheal intubation by EMS providers decreased, and the mechanical chest compression and response time increased. Although these variables were maximally adjusted between the epinephrine and norepinephrine groups in this study, it is highly likely that these factors contributed to the overall survival of OHCA patients in the region.

## 5. Conclusions

The introduction of prehospital epinephrine use in OHCA patients yielded no evidence of improvement in survival to discharge or favorable neurological outcomes and adversely affected the practice of EMS providers, exacerbating the factors negatively associated with survival of OHCA patients. Prehospital administration of epinephrine by EMS providers should be reconsidered.

## Figures and Tables

**Figure 1 jcm-11-00190-f001:**
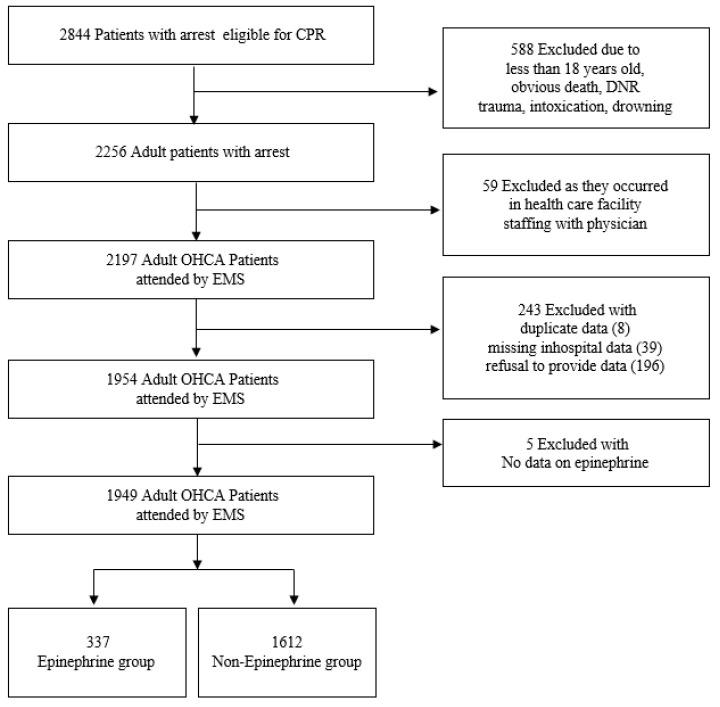
The flow chart of the study population. CPR, cardiopulmonary resuscitation; OHCA, out-of-hospital cardiac arrest; EMS, emergency medical service; DNR, do not resuscitate.

**Table 1 jcm-11-00190-t001:** Characteristics of the total study population and propensity-matched population according to prehospital epinephrine administration by EMS providers.

	Total Study Population			1:1 Propensity-Matched Population		
	Epinephrine Group (*n* = 337)	Norepinephrine Group (*n* = 1612)	*p*-Value	Epinephrine Group (*n* = 322)	Norepinephrine Group (*n* = 322)	*p*-Value
Patient variables						
Age (year) ^a^	69.87 ± 13.86	70.91 ± 15.24	0.057 ^1^	70.19 ± 13.71	70.51 ± 15.18	0.766 ^2^
	73 (62–80)	75 (61–82)		73 (62–80)	74 (61–81)	
Sex (male %)	227 (67.4)	993 (61.6)	0.047 ^3^	215 (66.8)	213 (66.2)	0.865 ^4^
Medical history						
Hypertension	99 (29.6)	437 (27.4)	0.402 ^3^	96 (29.8)	101 (31.4)	0.680 ^4^
Diabetes	65 (19.5)	298 (18.7)	0.737 ^3^	63 (19.6)	72 (22.4)	0.384 ^4^
Stroke	24 (7.2)	127 (8.0)	0.633 ^3^	24 (7.5)	21 (6.5)	0.647 ^4^
Cardiac disease	63 (18.9)	247 (15.5)	0.125 ^3^	61 (18.9)	69 (21.4)	0.424 ^4^
Pulmonary disease	25 (7.5)	116 (7.3)	0.890 ^3^	24 (7.5)	27 (8.4)	0.662 ^4^
Liver disease	10 (3.0)	21 (1.3)	0.027 ^3^	9 (2.8)	11 (3.4)	0.655 ^4^
Renal disease	13 (3.9)	65 (4.1)	0.879 ^3^	13 (4.0)	17 (5.3)	0.433 ^4^
Malignancy	27 (8.1)	170 (10.7)	0.159 ^3^	26 (8.1)	36 (11.2)	0.174 ^4^
Bystander variables						
Bystander witnessed	131 (38.9)	686 (42.6)	0.213 ^3^	125 (38.8)	112 (34.8)	0.293 ^4^
Bystander CPR	204 (60.5)	863 (53.5)	0.019 ^3^	195 (60.6)	199 (61.8)	0.746 ^4^
EMS variables						
Initial shockable rhythm						
Shockable (VF/pulseless VT)	45 (13.4)	152 (9.4)	0.030 ^3^	42 (13.0)	42 (13.0)	1.000^4^
Non-shockable (Asystole/PEA)	292 (86.7)	1460 (90.6)		280 (87.0)	280 (87.0)	
Advanced airway management						
I-gel/supraglottic airway	249 (73.9)	1159 (71.9)	0.294 ^3^	243 (75.5)	245 (76.1)	0.701 ^4^
Tracheal intubation	13 (3.9)	40 (2.5)		11 (3.4)	15 (4.7)	
No advanced airway	39 (11.6)	231 (14.3)		36 (11.2)	29 (9.0)	
No data	36 (10.7)	182 (11.3)		32 (9.9)	33 (10.3)	
Mechanical chest compression	223 (66.2)	519 (32.2)	<0.0001 ^3^	210 (65.2)	220 (68.3)	0.251 ^4^
EMS process time (minutes) ^b^						
(1) EMS response time	9.62 ± 5.78	8.88 ± 6.18	0.017 ^1^	9.46 ± 5.45	9.52 ± 7.76	0.416 ^5^
	8 (6–11)	7 (6–10)		8 (6–11)	8 (6–11)	
(2) EMS scene time	18.24 ± 5.87	13.57 ± 6.25	<0.0001 ^1^	17.99 ± 5.74	17.50 ± 6.14	0.068 ^5^
	18 (15–21)	13 (10–17)		18 (14–21)	17 (13–21)	
(3) EMS transport time	8.29 ± 9.29	9.67 ± 11.37	0.037 ^1^	8.40 ± 9.45	8.60 ± 8.13	0.504 ^5^
	6 (4–9)	6 (4–11)		6 (4–9)	6 (4–11)	

Variables are presented as the mean ± standard deviation ^a^, median (quartile 1–quartile 3) ^b^ and number (%). ^1^
*p*-values were derived from the Mann-Whitney U test, ^2^
*p*-values were derived from the paired *t*-test, ^3^
*p*-values were derived from the chi-square test, ^4^
*p*-values were derived from the McNemar test, and ^5^
*p*-values were derived from the Wilcoxon signed-rank test. EMS, emergency medical service; CPR, cardiopulmonary resuscitation; VF, ventricular fibrillation; VT, ventricular tachycardia; PEA, pulseless electrical activity; ROSC, return of spontaneous circulation.

**Table 2 jcm-11-00190-t002:** The effects of prehospital epinephrine administration by EMS providers during OHCA on survival to discharge and favorable neurological outcomes.

Variable		Total Study Population						Propensity-Matched Population (1:1)					
		Total	Survival	Incidence	*p*-Value ^1^	OR (95% CI)	*p*-Value ^2^	Total	Survival	Incidence	*p*-Value ^1^	OR (95% CI)	*p*-Value ^3^
ROSC at any time	Epinephrine group	337	89	26.4	0.290	0.87 (0.67~1.13)	0.290	322	85	26.4	0.233	1.25 (0.87~1.80)	0.230
	Norepinephrine group	1612	472	29.3		Reference		322	72	22.4		Reference	
Survival to discharge	Epinephrine group	337	17	5.0	0.215	0.72 (0.43~1.21)	0.217	322	15	4.7	0.699	1.15 (0.55~2.43)	0.706
	Norepinephrine group	1612	111	6.9		Reference		322	13	4.0		Reference	
Favorable neurologic outcomes	Epinephrine group	337	12	3.6	0.218	0.68 (0.37~1.26)	0.221	322	10	3.1	0.816	1.11 (0.45~2.73)	0.819
	Norepinephrine group	1612	83	5.2		Reference		322	9	2.8		Reference	

EMS, emergency medical service; OHCA, out-of-hospital cardiac arrest; ROSC, return of spontaneous circulation; OR, odds ratio; CI, confidence interval, ^1^ *p*-values were derived from the chi-square test, ^2^
*p*-values were derived from logistic regression analysis, ^3^
*p*-values were derived from conditional logistic regression analysis.

**Table 3 jcm-11-00190-t003:** The subgroup analysis according to the initial rhythm on the effect of prehospital epinephrine administration by EMS providers during OHCA on survival to discharge and favorable neurological outcomes.

Variable		1:1 Matched of Initial Non-Shockable Rhythm Group						1:1 Matched of Initial Shockable Rhythm Group					
		Total	Survival	Incidence	*p*-Value ^1^	OR (95%CI)	*p*-Value ^3^	Total	Survival	Incidence	*p*-Value ^1^	OR (95%CI)	*p*-Value ^3^
ROSC at any time	Epinephrine group	280	70	25.0	0.083	1.50 (0.96~2.3)	0.076	42	15	35.7	0.374	0.33 (0.04~3.21)	0.341
	Norepinephrine group	280	53	18.9				42	19	45.2			
Survival to discharge	Epinephrine group	280	5	1.8	1.000	1.00 (0.29~3.4)	1.000	42	10	23.8	0.595	2.00 (0.18~22.06)	0.571
	Norepinephrine group	280	5	1.8				42	8	19.1			
Favorable neurologic outcomes	Epinephrine group	280	1	0.4	1.000 ^2^	0.50 (0.05~5.5)	0.571	42	9	21.4	0.578	2.00 (0.18~22.06)	0.571
	Norepinephrine group	280	2	0.7				42	7	16.7			

^1^ *p*-values were derived from chi-square test; ^2^ *p*-values were derived from Fisher’s exact test; ^3^ *p*-values were derived from conditional logistic regression analysis.

**Table 4 jcm-11-00190-t004:** The prognostic factors affecting survival to discharge in prehospital epinephrine administration by EMS providers.

Variable	Univariate Analysis		Multivariate Analysis	
	OR (95% CI)	*p*-Value	OR (95% CI)	*p*-Value
Epinephrine group	1.16 (0.54~2.48)	0.699		
Patient variables				
Age (year)	0.96 (0.94~0.98)	<0.0001	0.97 (0.94~1.00)	0.028
Sex (male %)	3.15 (1.08~9.19)	0.036	1.41 (0.43~4.69)	0.574
Medical history				
Hypertension	0.48 (0.18~1.28)	0.143		
Diabetes	0.28 (0.07~1.19)	0.085		
Stroke	0.48 (0.06~3.63)	0.478		
Cardiac disease	2.70 (1.23~5.92)	0.013	2.39 (0.92~6.24)	0.075
Pulmonary disease	0.89 (0.21~3.86)	0.877		
Liver disease	1.16 (0.15~9.02)	0.885		
Renal disease	0.75 (0.10~5.71)	0.781		
Malignancy	0.34 (0.05~2.52)	0.290		
Bystander variables				
Bystander-witnessed	2.38 (1.11~5.13)	0.026	0.95 (0.37~2.45)	0.916
Bystander CPR	1.95 (0.82~4.67)	0.131		
EMS variables				
Initial rhythm				
Shockable (VF/pulselessVT)	15.00 (6.65~33.86)	<0.0001	8.86 (3.32~23.68)	<0.0001
Non-shockable (Asystole/PEA)	reference		reference	
Advanced airway management				
I-gel/supraglottic airway	0.40 (0.15~1.04)	0.645		
Tracheal intubation	0.82 (0.15~4.35)	0.375		
No data	0.15 (0.02~1.31)	0.153		
No advanced airway	reference		reference	
Mechanical chest compression	0.12 (0.05~0.31)	<0.0001	0.13 (0.05~0.36)	<0.0001
EMS process time(minutes)				
(1) EMS response time	1.00 (0.95~1.06)	0.925		
(2) EMS scene time	1.02 (0.96~1.09)	0.532		
(3) EMS transport time	1.03 (1.01~1.06)	0.015	1.02 (0.99~1.06)	0.145

EMS, emergency medical service; CPR, cardiopulmonary resuscitation; VF, ventricular fibrillation; VT, ventricular tachycardia; PEA, pulseless electrical activity; OR, odds ratio; CI, confidence interval.

**Table 5 jcm-11-00190-t005:** The prognostic factors affecting favorable neurological outcomes in prehospital epinephrine administration by EMS providers.

Variable	Univariate Analysis		Multivariate Analysis	
	OR (95% CI)	*p*-Value	OR (95% CI)	*p*-Value
Epinephrine group	1.12 (0.45~2.78)	0.816		
Patient variables				
Age (year)	0.94 (0.92~0.97)	<0.0001	0.95 (0.91~0.99)	0.012
Sex (male %)	9.44 (1.25~71.18)	0.029	3.02 (0.35~26.25)	0.316
Medical history				
Hypertension	0.42 (0.12~1.45)	0.168		
Diabetes	0.20 (0.03~1.54)	0.123		
Stroke	0.73 (0.10~5.62)	0.766		
Cardiac disease	2.38 (0.92~6.17)	0.074		
Pulmonary disease	0.64 (0.08~4.89)	0.666		
Liver disease	<0.001 (<0.001~>999.999) ^1^	0.982		
Renal disease	1.14 (0.15~8.85)	0.899		
Malignancy	0.51 (0.07~3.92)	0.520		
Bystander variables				
Bystander-witnessed	3.88 (1.45~10.35)	0.007	0.84 (0.23~3.14)	0.796
Bystander CPR	5.59 (1.28~24.41)	0.022	3.89 (0.73~20.74)	0.112
EMS variables				
Initial rhythm				
Shockable (VF/pulselessVT)	43.67 (12.41~153.70)	<0.0001	23.41 (5.59~98.10)	<0.0001
Non-shockable(Asystole/PEA)	Reference		reference	
Advanced airway management				
I-gel/supraglottic airway	0.45 (0.14~1.41)	0.969		
Tracheal intubation	<0.001 (<0.001~>999.999) ^1^	0.969		
No data	0.24 (0.03~2.19)	0.977		
No advanced airway	reference		reference	
Mechanical chest compression	0.13 (0.04~0.38)	<0.0001	0.15 (0.04~0.53)	0.003
EMS process time(minutes)				
(1) EMS response time	1.01 (0.96~1.07)	0.610		
(2) EMS scene time	1.02 (0.95~1.10)	0.559		
(3) EMS transport time	1.04 (1.01~1.07)	0.007	1.04 (1.00~1.08)	0.078

EMS, emergency medical service; CPR, cardiopulmonary resuscitation; VF, ventricular fibrillation; VT, ventricular tachycardia; PEA, pulseless electrical activity; OR, odds ratio; CI, confidence interval, ^1^ Odds ratio is not estimable since no patients with liver disease and with tracheal intubation experienced favorable neurologic outcomes.

## Data Availability

Raw data were generated at the national fire agencies in Korea. Derived data supporting the findings of this study are available from the corresponding author (SHK) on reasonable request.

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
