# Peer review of "Effect of Prehospital Epinephrine Use on Survival from Out-of-Hospital Cardiac Arrest and on Emergency Medical Services"

_jcm, 2021, doi:10.3390/jcm11010190_

Round 1

Reviewer 1 Report

Congratulation, it is very good prepared paper with appropriate methodology, statistical analysis and results presentation.

However I have few comments:

  1. authors should compare results with PARAMEDIC2 study in discussion.
  2. Most part of methods should be included to discussion: especially part of EMS organization in region
  3. the chapter"

    In this study, EMS providers used a mechanical chest compression device and performed manual chest compression more often in the epinephrine group (66.2% versus 305 32.2%, P<0.001). However, mechanical chest compression was a negative prognostic factor for survival to discharge (0.25, 014-0.44) and favorable neurological outcomes (0.20, 307 0.09-0.42) of OHCA patients in the epinephrine group. In this study, it is possible that mechanical chest compression was of lower quality than manual chest compression."

    despite statistical findings that comment is rather speculation. Please provide merit comments with citation.

    We know that in. eg. Lucas device provide very good quality chest compression with proper CCF. What kind of device was used?

    how looks the chest compression during transportation? It is safe?

    was used any device for chest compression control? any additional devices, capnometry, capnography?

Author Response

We thank reviewer for your insightful review, and time and effort spent on the same. We have tried to address all the suggestions as indicated in the review report.

However I have few comments:

  1. authors should compare results with PARAMEDIC2 study in discussion.

We thank the reviewer for this valuable comment, and we agree with the reviewer that the PARAMEDIC2 study is a very important study. Therefore, we have added the following to the discussion section of the revised manuscript.

A study by Perkins et al. showed that the use of epinephrine resulted in a significantly higher rate of 30-day survival than the use of placebo (3.2% in the epinephrine group and 2.4% in the placebo group, 1.39, 1.06-1.82). However, our study did not show any improvement in survival to discharge (5.0% in the epinephrine group and 6.9% in the nonepinephrine group, 0.72, 0.43-1.21) [7]. Factors contributing to such difference might include difference in the proportion of witness of cardiac arrest. The proportion of witnessed cardiac arrest in the previous study was 61.4% in the epinephrine group and 61.0% in the placebo group. However, in our study, it was 38.9% in the epinephrine group and 42.6% in the nonepinephrine group. This was because our study included patient cohort during the COVID-19 period. Another factor that might have contributed to different results between the present study and the PARAMEDIC2 study was that ALS was performed sufficiently in the scene, with a mean time interval between ambulance arrival at scent and departure of 50.1±21.8 minutes in the epinephrine group and 44.5±18.3 minutes in the placebo group. However, in our study, we recommended not to stay at the scene for more than 15 minutes. In addition, epinephrine was not used in the ambulance being transported.

  1. Most part of methods should be included to discussion: especially part of EMS organization in region

We thank the reviewer for pointing this out and we agree with the reviewer. Therefore, we have reorganized the method section according to the STROBE guideline.

  1. the chapter"

In this study, EMS providers used a mechanical chest compression device and performed manual chest compression more often in the epinephrine group (66.2% versus 305 32.2%, P<0.001). However, mechanical chest compression was a negative prognostic factor for survival to discharge (0.25, 014-0.44) and favorable neurological outcomes (0.20, 307 0.09-0.42) of OHCA patients in the epinephrine group. In this study, it is possible that mechanical chest compression was of lower quality than manual chest compression."

despite statistical findings that comment is rather speculation. Please provide merit comments with citation.

We know that in. eg. Lucas device provide very good quality chest compression with proper CCF. What kind of device was used?

how looks the chest compression during transportation? It is safe?

was used any device for chest compression control? any additional devices, capnometry, capnography?

Thank you for your critical comment. It is a very important point. We should have been a little more balanced in this part. Therefore, we modified it as follows and added two references.

In this study, EMS providers used a mechanical chest compression device more often than performed manual chest compression in the epinephrine group (66.2% versus 32.2%, P < 0.001). However, mechanical chest compression was a negative prognostic factor for survival to discharge (0.25, 014-0.44) and favorable neurological outcomes (0.20, 0.09-0.42) of OHCA patients in the epinephrine group. From these findings, we have two questions: 1) why mechanical chest compression devices were used more often in the epinephrine group than in the nonepinephrine group? and 2) was the quality of mechanical chest compression inferior to that of manual chest compression? Our estimation for the first issue was that with the current situation that three EMS providers were dispatched to one ambulance (one of them driving), the introduction of prehospital epinephrine use was believed to have caused a shortage of labor to apply manual chest compression, although EMS resuscitation protocol introduced multiple dispatches (two or more ambulance teams) as described in the study setting. Regarding the second question, EMS providers in the region use LUCAS, Easy Pulse, and AutoPulse as mechanical chest compression devices. However, equipment that can monitor and control the quality of CPR including capnometry and CPR quality improvement programs is not routinely used. In an animal study comparing LUCAS and manual chest compression, the use of mechanical devices provided ongoing high‐quality chest compression and tissue perfusion during ambulance transport [25]. Another study showed that patients in the mechanical CPR group had a higher chest compression fraction than those in the manual CPR group (0.84, 0.78-0.91) [26]. However, some studies have indicated no evidence that the use of mechanical chest compression devices is associated with improved survival rate of OHCA patients or poor neurological outcomes at hospital discharge [27-29]. The second question requires in-depth investigation and further research based on regional data.

Reviewer 2 Report

I really love this topic

The research was developed very well: an introduction is adequate

Methods are detailed and the choice of propensity score match is right

Results are clearly expressed: very interesting no-difference for age between groups

Discussion explore and discuss the data correctly

This study deserves publications

Author Response

We thank the reviewer for your positive comments regarding our manuscript.